# Multi-Track Timeline Control for Text-Driven 3D Human Motion Generation

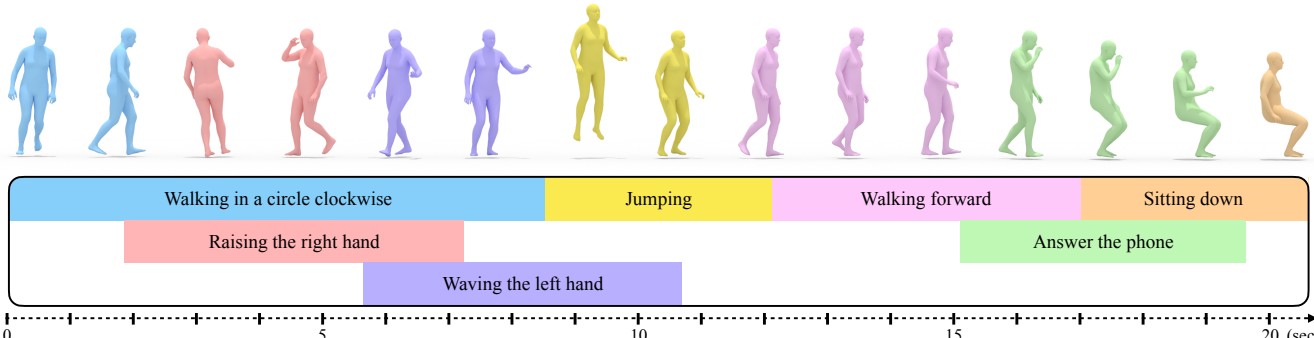

Figure 1. **Multi-track timeline control:** We introduce a new problem setting for text-driven motion synthesis, where the input consists of parallel tracks allowing simultaneous actions, as well as continuous temporal intervals enabling sequential actions. A long and complex motion can be generated (top) given the structured input of multiple simple textual descriptions, each corresponding to a temporal interval (bottom).

## Abstract

*Recent advances in generative modeling have led to promising progress on synthesizing 3D human motion from text, with methods that can generate character animations from short prompts and specified durations. However, using a single text prompt as input lacks the fine-grained control needed by animators, such as composing multiple actions and defining precise durations for parts of the motion. To address this, we introduce the new problem of timeline control for text-driven motion synthesis, which provides an intuitive, yet fine-grained, input interface for users. Instead of a single prompt, users can specify a multi-track timeline of multiple prompts organized in temporal intervals that may overlap. This enables specifying the exact timings of each action and composing multiple actions in sequence or at overlapping intervals. To generate composite animations from a multi-track timeline, we propose a new test-time denoising method. This method can be integrated with any pre-trained motion diffusion model to synthesize realistic motions that accurately reflect the timeline. At every step of denoising, our method processes each timeline interval (text prompt) individually, subsequently aggregating the predictions with consideration for the specific body parts engaged in each action. Experimental comparisons and ablations validate that our method produces realistic motions that respect the semantics and timing of given text prompts.*

## 1. Introduction

Motivated by applications in video games, entertainment, and virtual avatar creation, recent work has demonstrated substantial progress in learning to generate 3D human motion [27, 37, 44, 60]. Generating motions from text descriptions is of particular interest; it has the potential to democratize animation with a natural language interface that is intuitive for beginner and expert users alike. To this end, several methods have been proposed that synthesize reasonable character animations given a single text prompt and fixed duration as input [38, 53, 65].

While these methods are a promising first step towards faster and more accessible animation interfaces, they lack the precise control that is crucial for many animators. Consider the input prompt (see Fig. 2d): "A human walks in a circle clockwise, then sits, simultaneously raising their right hand towards the end of the walk, the hand raising halts midway through the sitting action." Due to a lack of representative training data, prior work struggles with such complex text prompts [38, 53]. Namely, the prompt includes *temporal* composition [4] where multiple actions are performed in sequence (e.g., walking *then* sitting), along with *spatial* composition [5] where several actions are performed simultaneously with differing body parts (e.g., walking *while* raising hand). Furthermore, such lengthy prompts quickly become unwieldy for the user and, despite their detailed descriptions, are still ambiguous with respect to the timing and duration of the constituent actions.

Figure 2. **Text-driven motion synthesis tasks:** Our framework generalizes (a) traditional *text-to-motion synthesis* given one text and one duration, (b) *temporal composition* given a sequence of texts for non-overlapping intervals, and (c) *spatial composition* given a set of texts for a single interval. (d) *Multi-track timeline control* uses a set of texts for arbitrary intervals, allowing fine-grained control over the timings of several complex actions.

To improve controllability, we propose the new problem of *multi-track timeline control for text-driven 3D human motion synthesis*. In this task, the user provides a structured and intuitive timeline as input (Fig. 1), which contains several (potentially overlapping) temporal intervals. Each interval corresponds to a precise textual description of a motion. As shown in Fig. 2d, the complex example prompt discussed earlier becomes simple to specify within the timeline, and allows animators to control the timing of each action. Such a timeline interface is already common in animation and video editing software, and is analogous to control interfaces that have recently emerged from the text-to-image community [64], e.g., image generation from a segmentation mask.

Multi-track timeline control for text-driven motion synthesis is a generalization of several motion synthesis tasks, and therefore brings many challenges. In particular, the multi-track timeline input can achieve (see Fig. 2):

- *Text-to-motion synthesis* [18, 38] – specifying a single interval (i.e., duration) with one textual description,
- *Temporal composition* [4, 66] – a sequence of textual descriptions corresponding to non-overlapping intervals,
- *Spatial (body-part) composition* [5] – a set of text prompts performed simultaneously with differing body parts.

Solving this task is difficult due to the lack of training data containing complex compositions and long durations. For example, a timeline-controlled model must handle the multi-track input containing several prompts, rather than a single text description. Moreover, the model must account for both spatial and temporal compositions to ensure seamless transitions, unlike prior work that has addressed each of these individually. The timeline also relaxes the assumption of a limited duration ($<$10 sec) made by many recent text-to-motion approaches [11, 53, 65].

To address these challenges, we introduce a method for **S**patio-**T**emporal **M**otion **C**ollage (STMC). Our method copes with the lack of appropriate training data by operating at test time, leveraging a pre-trained motion diffusion model such as off-the-shelf MDM [53] or MotionDiffuse [65]. At each denoising step, STMC first applies the diffusion model on each text prompt in the timeline independently to predict a denoised motion for the corresponding intervals. Our key insight is to stitch together such independent generations in both space and time before continuing to denoise. For spatial compositions, automatic body part associations [5] allow coherently concatenating predictions together. Score arithmetic [66] is used to ensure smooth transitions for temporal compositions. To further improve the performance of STMC, we introduce MDM-SMPL, which makes several improvements to prior motion diffusion models [53], including directly using the SMPL [34] body representation.

The performance of STMC on timeline control for text-driven motion synthesis is verified through comprehensive comparisons and a perceptual user study. In summary, the central contribution of this work consists of: (i) the new problem of *multi-track timeline control for text-driven 3D human motion synthesis*, and (ii) a novel test-time technique, STMC, that effectively structures the denoising process to ensure faithful execution of all prompts in a timeline. As a side contribution, (iii) we upgrade MDM to directly support the SMPL body representation instead of skeletons, and reduce runtime through fewer denoising steps. Code will be released upon publication.

## 2. Related Work

**Human motion synthesis**. A large body of work in both vision and graphics has been dedicated to generating 3D human motions [70]. This generation process can be unconditional [36, 56] or conditioned on actions [10, 17, 37], music [32, 50, 52, 57], speech [3, 69], goals [30, 51, 60], previous motion [13, 15, 44, 62] (i.e., future motion prediction), scenes/objects [21, 31, 58, 59], and text [1, 2, 11, 16, 19, 29, 53, 65]. Technical approaches vary from early statistical models [8, 15] to modern generative models like VAEs [20, 37, 38], GANs [6, 12, 49, 61], normalizing flows [22, 57], and diffusion [11, 29, 30, 60, 68]. Our work is most related to recent text-conditioned diffusion models [53, 65], however we solve a new problem where the model is conditioned on a timeline containing several text inputs instead of a single prompt.

**Motion composition**. Due to the lack of training data, a particular challenge for action and text-conditioned motion generation is to synthesize compositional motions. Several works [4, 41, 66] focus on generating motions from a sequence of text prompts and durations, i.e., *temporal* compositions. TEACH [4] autoregressively generates one motion (per text prompt) at a time, conditioning the next motion in the sequence with the previous one. EMS [41] proposes a two-stage approach, by first generating each action separately and then merging them through a subsequent network. Diffusion models EDGE [54] and PriorMDM [48] ensure consistency between adjacent motions by enforcing temporal constraints at transitions. Our approach to temporal composition is based on DiffCollage [66], which stitches motions (or images) together throughout the denoising process via score arithmetic at overlapping transitions.

Other work generates motions from a set of texts to be executed at the same time, i.e., *spatial* (body-part) composition.

SINC [5] labels ground truth motion capture (mocap) sequences with corresponding body parts by prompting GPT-3 [9]. These labels are used to create a synthetic dataset of motions stitched together from mocap sequences with compatible body parts, thereby improving performance of VAE-based 3D motion generation methods [38] for spatial composition. MotionDiffuse [65] proposes a noise interpolation method to control different body part motions separately. Our approach, STMC, takes inspiration from SINC [5] by using body part labels to stitch motions together during test-time denoising. Overall, our problem of timeline-conditioned generation generalizes temporal and spatial composition, and STMC must tackle both issues simultaneously, unlike most prior work.

**Controllable motion diffusion**. Following success in image [43, 46, 47], video [26], and 3D [33, 40, 63] domains, diffusion has become a useful approach to generate high-quality 3D human motions [3, 28, 54], especially from text inputs [11, 14, 53, 65]. Some works focus on improving the controllability of motion diffusion models, e.g., by enabling temporal [48, 66] and spatial [65] composition of text prompts. Other controls such as following specific keyframe poses, joint trajectories, and waypoints have also been achieved using a mix of test-time diffusion guidance [28, 30, 45], in-painting [48, 53], and direct conditioning [60]. We focus on making text-to-motion generation more controllable by handling several text prompts in a fine-grained timeline format through a compositional denoising process.

## 3. Human Motion Synthesis from Timelines

We first formulate the new problem setup of multi-track timeline control (Sec. 3.1), then propose a motion denoising strategy to handle timeline inputs (Sec. 3.2 and Sec. 3.3), and finally summarize our improved diffusion model (Sec. 3.4).

### 3.1. Timeline Control Problem Formulation

**Inputs**. As illustrated in Fig. 1, the multi-track timeline enables users to define multiple intervals, each linked to a natural language prompt describing the desired human motion. For the $j$th prompt in the timeline, we represent its temporal interval as $[a_j, b_j]$ and the corresponding prompt as $C_j$. The intervals are arranged in a multi-track layout on the timeline, allowing for overlaps. Both the duration of each interval and of the overall timeline are variable, and users can add an arbitrary number of tracks (rows) to the timeline (although, in practice, a character can most often perform a handful of actions simultaneously).

**Outputs**. The goal is to generate a 3D human motion that follows all the text instructions at the specified intervals. A human motion $x$ lasting $N$ timesteps is represented as a sequence of pose vectors $x = (x^1, ..., x^N)$ with each pose $x^i \in \mathbb{R}^d$. Several recent works [53, 65] use the pose representation from Guo *et al.* [18] with $d = 263$, which contains root velocities along with local joint positions, rotations, and velocities. Other pose representations like SMPL [34] can also be used (see Sec. 3.4).

## 3.2. Background: Motion Diffusion Models

Our generation method (Sec. 3.3) leverages a pre-trained motion diffusion model such as MDM [53] or MotionDiffuse [65] trained on single text prompts, which we briefly review here. These methods follow a denoising diffusion scheme and synthesize animations through iterative denoising of a noisy pose sequence. Given a clean motion $x_0$, a Gaussian diffusion process is employed to corrupt the data to be approximately $\mathcal{N}(\mathbf{0}, \mathbf{I})$. Each step of this process is given by:

$$q(x_t | x_{t-1}) = \mathcal{N}(x_t; \sqrt{1 - \beta_t} x_{t-1}, \beta_t \mathbf{I}) \tag{1}$$

with $\beta_t$ defined by the noise schedule. Note the denoising step $t$ is not to be confused with the temporal timestep $i$, which indexes the sequence of poses in the motion. In practice, one can make sampling $x_t$ easier by using the reparameterization trick $x_t = \sqrt{\bar{\alpha}_t} x_0 + \sqrt{1 - \bar{\alpha}_t} \epsilon$, where $\epsilon \sim \mathcal{N}(\mathbf{0}, \mathbf{I})$, $\alpha_t = 1 - \beta_t$, and $\bar{\alpha}_t = \prod_{s=0}^{t} \alpha_s$.

Sampling from a diffusion model requires reversing this process to recover a clean motion from random noise. While $q(x_{t-1} | x_t)$ is hard to compute, the probability conditioned on $x_0$ is tractable [25]:

$$q(x_{t-1} | x_t, x_0) = \mathcal{N}(x_{t-1}; \mu_t(x_t, x_0), \mathbf{\Sigma}_t), \tag{2}$$

where

$$\mu_t(x_t, x_0) = \frac{\sqrt{\alpha_t}(1 - \bar{\alpha}_{t-1})}{1 - \bar{\alpha}_t} x_t + \frac{\sqrt{\bar{\alpha}_{t-1}} \beta_t}{1 - \bar{\alpha}_t} x_0 \tag{3}$$

$$\mathbf{\Sigma}_t = \frac{1 - \bar{\alpha}_{t-1}}{1 - \bar{\alpha}_t} \beta_t \mathbf{I}. \tag{4}$$

Since $x_t$ is known at sampling time, we approximate the reverse distribution by training a denoising model $\hat{x}_\theta(x_t, t, C)$ to estimate $x_0$, where $C$ is the text conditioning. This model is trained with the simplified loss function as in Ho et al. [25] (i.e., without the $t$-dependent factor):

$$\mathcal{L} = \mathbb{E}_{\epsilon, t, x_0, C} \| \hat{x}_\theta(x_t, t, C) - x_0 \|_2^2 \tag{5}$$

with $x_0$ and $C$ sampled from a dataset of motion-text pairs, step $t$ sampled uniformly, and noise $\epsilon \sim \mathcal{N}(\mathbf{0}, \mathbf{I})$ used to corrupt the ground truth motion. To enable classifier-free guidance [24] at sampling time, the text conditioning $C$ is dropped with some probability at each training iteration. At test time, the sampling (reverse) process starts from random noise and denoises iteratively for $T$ steps to obtain a clean 3D human motion. At each denoising step, the model is conditioned on the single input text prompt (e.g., Fig. 2a).

## 3.3. STMC: Spatio-Temporal Motion Collage

STMC operates only at test time, enabling an off-the-shelf, pre-trained denoising model to generate motion conditioned on a multi-track timeline. At *every* denoising step, our method takes as input the current noisy motion $x_t$ encapsulating the entire

Figure 3. **Overview of STMC:** Before denoising, the multi-track timeline is first **(a)** partitioned into relevant body parts per text (using LLM-based labeling [5]) to create body part timelines, which are then **(b)** extended to overlap, leading to the transition intervals used for temporal stitching *per body part* with DiffCollage [66]. **(c)** At each denoising step, motions for each prompt are denoised independently before being combined based on the body-part timelines. The composite motion is re-noised by sampling $x_{t-1}$ from $\mathcal{N}(\mu_t(x_t, \hat{x}_0), \Sigma_t)$ (as in Eq. (2)) before being passed to the next step.

timeline and outputs a corresponding clean motion $\hat{x}_0$. As shown in Fig. 3c, STMC uses the denoising model to independently predict a clean motion crop corresponding to each of the input text prompts. These predictions are stitched together spatially using body part annotations for each text prompt (Fig. 3a), and stitched in time to ensure the clean motion smoothly spans the entire timeline (Fig. 3b). This final composite motion becomes the output of the current step $\hat{x}_0$, which is used to sample $x_{t-1}$ with Eq. (2) and continue the denoising process. To enable body part stitching, STMC assumes the denoiser operates on explicit poses [53, 65], rather than in a latent space [11].

**Motion cropping and denoising**. The input $x_t$ at denoising step $t$ extends over the duration of the entire timeline. As shown in Fig. 3c, we first temporally split the input into motion "crops" to separately denoise each text prompt. For each interval $[a_j, b_j]$, the motion is cropped in time to $x_t^{a_j:b_j} = x_t[a_j : b_j]$. The crop, along with the text prompt $C_j$, is given to the denoising model to predict a corresponding clean motion crop $\hat{x}_0^{a_j:b_j}$. Denoising each text prompt independently gives high-quality motion from pre-trained models since each prompt typically contains a single action and the interval duration is reasonably short ($<$10 sec).

Two or more text prompts in the timeline may overlap in time, meaning the predicted clean crops will also overlap. As a concrete example, suppose the crops for "walking in a circle" and "raising right hand" are overlapping, as in Fig. 3. In this case, it is not clear which of the two generated motions should be assigned to the overlapping region. To construct a motion that matches both prompts, we need the leg motion from "walking in a circle" and the right arm motion from "raising right hand". We therefore stitch together outputs from overlapping prompts based on automatically labeled body parts, as detailed next.

**Spatial (body-part) stitching**. Spatial stitching follows SINC [5], which proposed to combine compatible body-

motions from mocap sequences through simple concatenation. While SINC applies stitching only once, STMC does so at *every* step of denoising, encouraging a more coherent composition of movements by allowing the denoiser to correct any artifacts. This is possible because the denoiser outputs explicit human poses (i.e., we know which indices correspond to arms, legs, etc. within the pose vector), so we can extract body-part motions from separate crops and spatially combine them to obtain a composite motion. To achieve this, we first pre-process the input timeline to assign a text prompt to each body part at every timestep, thereby creating a separate motion timeline for every body part (see Fig. 3a): *left arm*, *right arm*, *torso*, *legs* and *head*.

As shown in Fig. 3a, each text prompt in the multi-track timeline is first annotated with a set of body parts involved in the motion. This can be done automatically by querying GPT-3 [9] as in SINC, or directly given by the user for additional creative control. Then, each text prompt is assigned to its annotated body parts within the corresponding time interval, which assumes that body parts at overlapping intervals are compatible (e.g., if a prompt is annotated with "legs", then no other prompt should involve legs throughout its entire interval). To fill in the remainder of the body-part timelines where body parts have not been annotated to a text prompt, heuristics similar to SINC are used. Please see the Appendix B and the Fig. A.1 for full details. Finally, during the denoising step (Fig. 3c), each crop $x_t^{a_j:b_j}$ is split into separated body-part motions and concatenated together as specified by the body-part timelines to obtain the output $\hat{x}_0$.

**Temporal stitching**. Because the motion crops are denoised independently, simple temporal concatenation of body-part motions from different text prompts will cause abrupt transitions. To mitigate these potential artifacts, we apply DiffCollage [66] to *each body-part* motion. As shown in Fig. 3b, instead of directly denoising $x_t^{a_j:b_j}$ for each text prompt, we denoise an expanded time interval $[a_j - l, b_j + l]$, where $l$ is the desired over-

lap length between adjacent motion crops (e.g., fixed to 0.25 sec). Concretely, for the temporal transition between prompts $j$ and $k$, we have $\hat{x}_0^{a_j-l:b_j+l}$ and $\hat{x}_0^{a_k-l:b_k+l}$ after denoising. We then *unconditionally* denoise a small (0.5 sec) crop of motion centered on the overlap between $j$ and $k$ to obtain $\hat{x}_0^{\text{uncond}}$. The final predicted motion spanning intervals $j$ and $k$ is computed as $\hat{x}_0 = \hat{x}_0^{a_j-l:b_j+l} + \hat{x}_0^{a_k-l:b_k+l} - \hat{x}_0^{\text{uncond}}$, as depicted in Fig. 3c. This equation derives from a factor graph representation of the problem, as detailed in DiffCollage [66].

### 3.4. SMPL Support for Motion Diffusion Model

While STMC works well with off-the-shelf models [53, 65] (see Sec. 4), we propose several practical improvements to MDM [53] to further enhance results. Our model, MDM-SMPL, employs a skinned human body SMPL [34]: we use SMPL pose parameters instead of the joint rotation features in the original pose representation of Guo et al. [18]. In contrast to models that use the joint position outputs from the pose representation of [18], this SMPL-based representation avoids the need for expensive test-time optimization [7, 71] to fit the generated motion on a SMPL body. Moreover, the local joint rotations in SMPL, which are relative to parents in the kinematic tree, are more amenable to body-part stitching than root-relative joint positions. This is because any change to a joint rotation is propagated to all children in the kinematic tree, unlike root-relative joint positions which may not be coherent when simply concatenated together. Additional improvements include lowering the number of diffusion steps to $T=100$ from 1000 to substantially speed up sampling, and various architectural changes. We provide more details on MDM-SMPL in Appendix D.

## 4. Experiments

We first present the data (Sec. 4.1) and the evaluation protocols (Sec. 4.2) used in the experiments. We then show comparisons with baselines quantitatively (Sec. 4.3) and with a perceptual study (Sec. 4.4), followed by qualitative results (Sec. 4.5). We conclude with a discussion of the limitations (Sec. 4.6).

### 4.1. Datasets

**HumanML3D [18]** is a text-motion dataset that provides textual descriptions for a subset of the AMASS [35] and Human-Act12 [17] motion capture datasets. It consists of 44970 text annotations for 14616 motions. This dataset is used to train all diffusion models used in our experiments. For MDM [53] and MotionDiffuse [65], we use publicly available models pre-trained on the released version of HumanML3D with the original motion representation from Guo et al. [18]. Consequently, these methods require test-time optimization to obtain SMPL pose parameter outputs. For training our MDM-SMPL diffusion model, which is designed to directly generate SMPL pose parameters, we re-process the dataset and exclude the Human-Act12 subset as SMPL poses are not available for this dataset.

**Multi-track timeline (MTT) dataset**. To properly evaluate our new task, we introduce a new challenging dataset of 500 multi-track timelines. Each timeline in the dataset is automatically constructed and contains three prompts on a two-track timeline (e.g., Fig. 2d). To construct these timelines, we first manually collect a set of 60 texts covering a diverse set of "atomic" actions (e.g., "punch with the right hand", "jump forward", "run backwards", see Appendix C for the full list), and annotate the involved body parts for each text. To serve as ground truth for computing evaluation metrics (Sec. 4.2), we also select motion samples from AMASS that correspond to each text. Based on the atomic texts, we automatically generate timelines containing three prompts and two tracks (rows). For each timeline, the first track is filled with two consecutive prompts sampled from the set of texts and given randomized durations. A third random text with complementary body-part annotations is then placed in the second track at a random location in time.

The main reasons for restricting the evaluation to three prompts are (i) to keep the cognitive load for users low in the perceptual study, subsequently increasing the reliability of the results, and (ii) to construct a minimal setup where we can fairly compare against baselines in a controlled setting, eliminating confounding factors such as the number of prompts. Though these timelines contain only three prompts, they already pose a significant challenge (see Sec. 4.3). Examples of timelines in the dataset are provided in Fig. A.2 and qualitative results beyond three prompts can be found in the supplementary video.

### 4.2. Evaluation Metrics

Given the novelty of the task, identifying relevant metrics to evaluate different methods is crucial. Instead of relying on a single metric, we disentangle the evaluation of semantic correctness (how faithful individual motion crops are to the textual descriptions) from that of realism (e.g., temporal smoothness). **Semantic metrics**. Firstly, we evaluate the alignment between the generated motion and the text description within the specified intervals on the timeline, which we term "per-crop semantic correctness". To assess this, we utilize the recent text-to-motion retrieval model TMR [39]. Similar to how CLIP [42] functions for images and texts, TMR provides a joint embedding space that can be used to determine the similarity between a text and motion. Using TMR, we encode each atomic text prompt and corresponding motion from our MTT dataset to obtain ground truth text and motion embeddings, respectively. Each generated motion crop is also embedded and the *TMR-Score*, a measure of cosine similarity ranging from 0 to 1, is calculated between the generated motion embedding and the ground truth. We report both motion-to-text similarity by comparing against the ground truth text embedding (*TMR-Score M2T*) and motion-to-motion similarity against the ground truth motion embedding (*TMR-Score M2M*). Such embedding similarity measures are akin to BERT-Score [67] for text-text, CLIP-Score [23] for image-text, and more recently TEMOS-Score [4] for motion-motion similar-

ity. Since TMR is trained contrastively, its retrieval performance is better than TEMOS [38] which only trains with positive pairs, leading to our decision to instead use TMR-Score. Moreover, its embedding space is optimized with cosine similarity, making the values potentially more calibrated across samples.

Ideally, the *TMR-Score M2T* between a generated motion crop and the corresponding input text prompt should surpass those of other texts. Hence, we also measure motion-to-text retrieval metrics (as in [18]) including the frequency of the correct text prompt being in the top-1 (*R@1*) and top-3 (*R@3*) retrieved texts from the entire set of atomic texts.

**Realism metrics**. Secondly, we evaluate the realism of the generated motions, which includes transitioning smoothly between actions. While the Frechet Inception Distance (*FID*) between generated and ground truth motion in a learned feature space (e.g., TMR) is a common metric for quality, the embedding space of TMR is not trained on motions that are longer than 10 sec, and may therefore be unreliable for longer motions. Hence, we follow DiffCollage [66] and compute the *FID+* to evaluate transitions. The *FID+* metric measures FID based on 5 random 5-second motion crops from each timeline-conditioned motion generation. Following TEACH [4], we also measure the *transition distance* as the Euclidean distance (in cm) between the poses in two consecutive frames around the transition time. We choose to compute this distance in the local coordinate system of the body to more effectively capture transitions for individual body parts, rather than being dominated by global motion. This metric is sensitive to abrupt pose changes, and a motion should not have high transition distance to remain realistic.

**Perceptual study**. Since no quantitative metric can fully capture the subtleties of human motion, we also conduct perceptual studies, where human raters on Amazon Mechanical Turk judge the quality of the generated motions [55]. To compare two generation methods, raters are presented with two videos of generated motions side-by-side rendered on a skeleton. The multi-track timeline is also visible with an animated bar that progresses along the timeline as the videos play. Users are asked which motion is *more realistic* and which one is *better at following the text in the timeline*; they may choose one of the two motions or mark "no preference". The studies presented in Sec. 4.3 are performed on a set of 100 motions with multiple raters judging each pair. The preference for each video is determined by a majority vote from all raters. Responses are filtered for quality by using three "warmup" questions at the start of each 15-question survey along with two "honeypot" examples with objectively correct answers. The honeypot examples test a rater's understanding of the task: one example shows a motion with obviously severe limb stretching (realism understanding test) and the other displays a motion generated from a different timeline than the one displayed (timeline understanding test). If a rater fails to answer either of these questions correctly, all of their responses are discarded.

## 4.3. Quantitative Comparison with Baselines

We apply our STMC test-time approach on the pretrained diffusion models of MotionDiffuse [65], MDM [53], and MDM-SMPL (ours). For each denoiser, we establish several strong baselines by repurposing existing methods to the timeline-conditioned generation task for comparison. Results are shown in Tab. 1. Next to each method, the table indicates how many tracks the input timelines have (*#tracks*) and how many text prompts can be contained in a track (*#crops*). Next, we introduce each baseline and analyze results.

**Single-text input [53, 65] baseline**. The simplest approach to condition motion diffusion on a timeline is to convert the timeline into a single text description, which aligns with the model's training input format (e.g., Fig. 2a). Given that our timeline dataset is consistently comprised of three motions (A, B, and C), we formulate single-text prompts as follows: "A and then B while C". While timing information can be included in the prompt, e.g., "A for 4 seconds", this is out-of-distribution for models trained on HumanML3D, leading to worse results. This method parallels the baseline strategies of SINC [5] for spatial composition and TEACH [4] for temporal composition.

As shown for each denoiser in Tab. 1, this approach is ineffective for both semantic correctness metrics and realism. Since these models cannot generate motions longer than 10 sec and there is no timing information in the prompt, for this experiment, outputs are limited to a maximum duration of 10 sec and semantic correctness metrics are reported over the entire duration of the motion rather than per-crop. The poor performance is a result of the models not being trained on the types of complex compositional prompts that result from collapsing the timeline to a single text description.

**DiffCollage [66] baseline**. Instead of converting the multi-track timeline into a single prompt, one can collapse it into a single track timeline containing a series of consecutive text prompts, i.e., transform the problem to be one of temporal composition. DiffCollage can then be used to temporally compose the sequence of actions. For example, the timeline in Fig. 2d would be split into ["walking in a circle," "walking in a circle while raising the right hand," "sitting down while raising the right hand," "sitting down"]. Note that, unlike the single-text baseline, this splitting preserves the timings (*#crops*) in the timeline.

While the DiffCollage baseline generally produces smooth transitions and reasonable FID scores, the semantic accuracy is consistently worse than STMC. This is due to the complex spatial compositions within the prompts after collapsing the timeline into a single track, which models trained on HumanML3D struggle with. In contrast, STMC uses body-part stitching throughout denoising to compose actions from simpler prompts.

**SINC [5] baseline**. Rather than performing body-part stitching iteratively at every denoising step, an alternative approach is to stitch body motions together only once after all crops have finished the entire denoising process. This is most similar to SINC and forms the basis for two baselines that accept the full

| Method | Input type | | Per-crop semantic correctness | | | | Realism | |
| | #tracks | #crops | R@1 ↑ | R@3 ↑ | TMR-Score ↑ | | FID ↓ | Transition distance ↓ |
| | | | | | M2T | M2M | | |
| **Ground truth** | - | - | 55.0 | 73.3 | 0.748 | 1.000 | 0.000 | 1.5 |
| **MotionDiffuse [65]** | Single | Single | 10.9 | 21.3 | 0.558 | 0.546 | 0.621 | 1.9 |
| DiffCollage | Single | Multi | 22.6 | 43.3 | 0.633 | 0.612 | 0.532 | 4.6 |
| SINC w/o Lerp | Multi | Multi | 23.8 | 45.9 | 0.656 | 0.630 | 0.554 | 3.8 |
| SINC w/ Lerp | " | " | 24.9 | 46.7 | 0.663 | 0.632 | 0.552 | 1.0 |
| STMC (ours) | " | " | 24.8 | 46.7 | 0.660 | 0.632 | 0.531 | 1.5 |
| **MDM [53]** | Single | Single | 9.5 | 19.7 | 0.556 | 0.549 | 0.666 | 2.5 |
| DiffCollage | Single | Multi | 24.9 | 42.3 | 0.636 | 0.623 | 0.600 | 2.2 |
| SINC w/o Lerp | Multi | Multi | 21.5 | 41.8 | 0.629 | 0.626 | 0.638 | 10.2 |
| SINC w/ Lerp | " | " | 23.3 | 43.1 | 0.634 | 0.628 | 0.630 | 2.8 |
| STMC (ours) | " | " | 25.1 | 46.0 | 0.641 | 0.633 | 0.606 | 2.4 |
| **MDM-SMPL** | Single | Single | 12.1 | 23.5 | 0.573 | 0.578 | 0.484 | 1.8 |
| DiffCollage | Single | Multi | 29.1 | 49.7 | 0.675 | 0.656 | 0.446 | 1.2 |
| SINC w/o Lerp | Multi | Multi | 32.3 | 50.5 | 0.676 | 0.667 | 0.463 | 4.2 |
| SINC w/ Lerp | " | " | 31.8 | 51.0 | 0.679 | 0.668 | 0.457 | 1.2 |
| STMC (ours) | " | " | 30.5 | 50.9 | 0.675 | 0.665 | 0.459 | 0.9 |

Table 1. **Quantitative baseline comparison**: Our method STMC is compared to several strong baselines when using three different denoising models. The single-text and DiffCollage baselines struggle to handle complex compositional prompts that results from collapsing the timeline down to a single track. The SINC baselines produce reasonable semantic accuracy by denoising prompts independently as in STMC, but cause abrupt or unnatural transitions with higher transition distance (underlined) or FID.

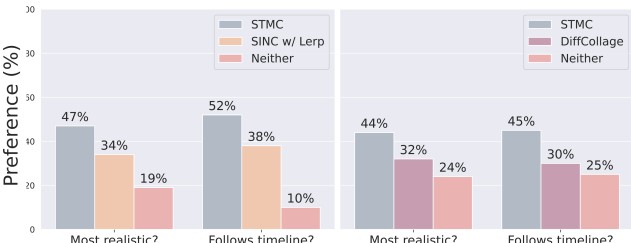

Figure 4. **Perception study results:** Our STMC method is preferred over baselines by human raters for both motion realism and semantic accuracy. (Left) Comparison against the strong SINC with Lerp baseline. (Right) Comparison against the DiffCollage baseline. MDM [53] is used as the denoiser in these experiments.

multi-track timeline as input, similar to STMC.

*SINC w/o Lerp* concatenates body part motions at the end of denoising without considering temporal transitions. As a result, transitions tend to be abrupt as evidenced by high transition distances in Tab. 1 and occasional "teleporting" limbs in qualitative results. To mitigate this, *SINC w/ Lerp* employs linear interpolation (lerp) at transitions for smoother results, similar to the approach in TEACH [4]. Though this leads to smoothness at transitions, FID scores tend to be slightly higher than STMC. The cause is obvious qualitatively, where the generated motion often appears mechanical and unnatural, sometimes resulting in foot sliding. Despite issues with motion quality, these SINC baselines effectively capture the semantics of each motion crop since crops are denoised independently.

**Analysis of the results**. Our method STMC consistently per-

forms effectively across *both* semantic and realism metrics, unlike baselines that tend to sacrifice performance in one category for the other. For example, DiffCollage achieves the best FID using MDM, but its inability to handle spatial compositions results in worse semantics than STMC across all models. Additionally, SINC baselines perform best in terms of semantics for MotionDiffuse and MDM-SMPL, but result in abrupt or unnatural transitions with FID or transition distance that is often higher than STMC. Such transitions are also readily apparent in qualitative results (see supplementary video). It is also notable that using MDM-SMPL with STMC performs on par with MDM and MotionDiffuse, while enabling direct SMPL output and significantly reducing (by 10×) the number of diffusion steps. Fewer steps, combined with pre-computing text embeddings, enable sampling MDM-SMPL in less than 5 seconds on average. This is a substantial improvement over MDM, which takes 4 minutes to generate motions followed by 8 min of optimization to obtain SMPL poses, on average.

While the performance of STMC is promising, the semantic metrics for ground truth motions indicate room for improvement. As discussed in Sec. 4.6, STMC is currently limited by the pre-trained diffusion model that it leverages for each motion crop; we expect improvements in these models to also boost STMC. An additional experiment on varying the overlap length for temporal stitching can be found in Appendix E.

## 4.4. Perceptual Study

We perform two separate user studies to compare STMC to *SINC with Lerp* and *DiffCollage* when using MDM. Fig. 4

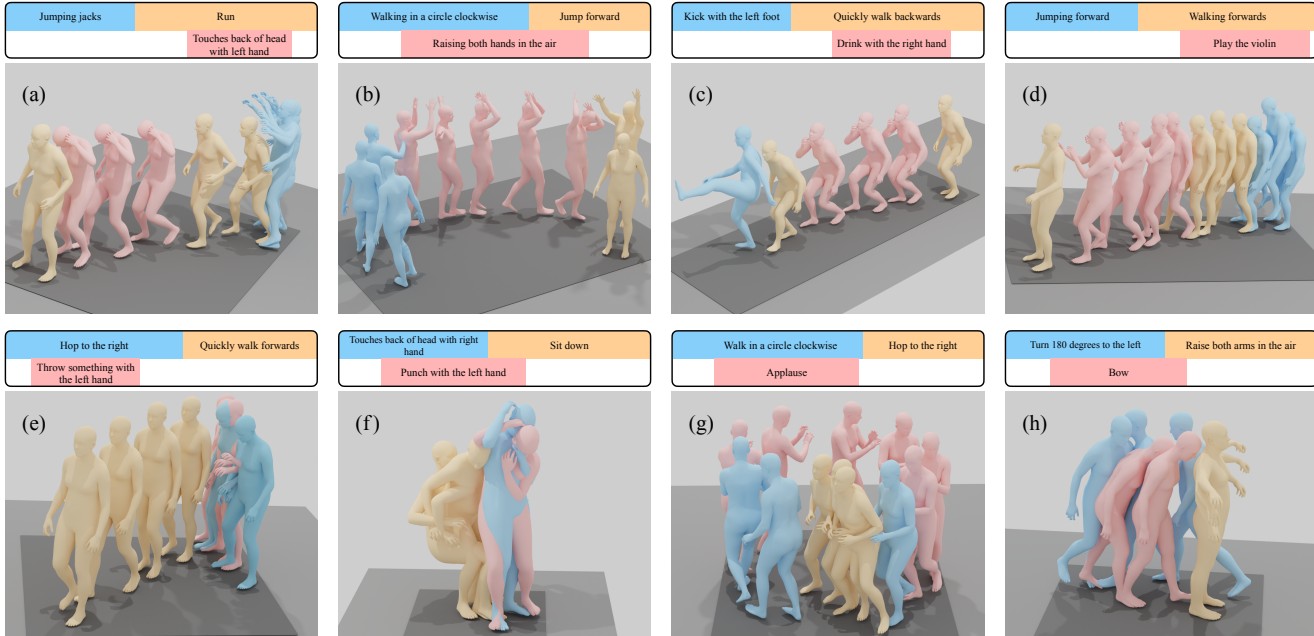

Figure 5. **Qualitative results:** We visualize the results of STMC with MDM-SMPL on several input timelines and color the bodies depending on their location in the timeline. We see that STMC is capable of generating realistic motions, which capture the semantics of the given text prompts with the desired timing and duration. In **(a)** and **(c)**, STMC generates motions that precisely follow the instructions, controlling a single arm while still performing another action. The accurate timing of intervals is demonstrated in **(b)** where the arms are still up in the air when transitioning from "walking" to "jumping", which is difficult to achieve with alternative methods. In **(c)** and **(d)**, we observe that STMC is capable of generating compositions that were not present in the ground truth data, such as "walking backwards while eating" or "walking while playing violin".

shows results of both studies, measuring human preference for motion realism and semantic accuracy. On the left, STMC is preferred or similar to SINC 66% of the time for realism and 62% of the time for semantic accuracy, with 4.2 raters judging each video on average after filtering bad responses. Compared to DiffCollage on the right, our method is preferred or similar 68% of the time for realism and 70% for semantic accuracy, with 2.8 raters judging each video after filtering. This demonstrates that STMC improves the motion in ways that are discernible by humans but may not be fully captured in quantitative metrics.

### 4.5. Qualitative Results

We visualize motions generated by STMC with MDM-SMPL in Figure 5, given multi-track timelines as input from our MTT dataset. The coloring follows the input text, prioritizing the newest prompt when there is an overlap across tracks. These results show that STMC is capable of generating realistic motions for complex multi-prompt timelines, which follow the timing and duration of the given intervals. Please see the caption for full analysis of these examples, and we refer to the supplementary video for additional qualitative results and comparison to generated motions from baseline methods.

### 4.6. Limitations

While STMC expands the capabilities of pre-trained motion diffusion models to take a multi-track timeline as input, it is also limited by the models that it relies on. For example, our proposed body-part stitching process produces spatially composed motions throughout denoising that the off-the-shelf models are not trained to robustly handle. One potential direction to ameliorate this is a more sophisticated stitching "schedule" where body parts are not combined until later in the denoising process instead of at every step. STMC also inherits the limitations of SINC, e.g., restricting overlapping motions to have compatible body part combinations.

## 5. Conclusion

In this work, we proposed the new problem of multi-track timeline control for text-driven 3D human motion generation. The timeline input gives users fine-grained control over the timing and duration of actions, while still maintaining the simplicity of natural language. We tackled this challenging problem using a new test-time denoising process called spatio-temporal motion collage (STMC), which enables pre-trained diffusion models to handle the spatial and temporal compositions present in timelines. Finally, extensive quantitative and qualitative evaluation demonstrated the advantage of STMC over strong baseline methods and its ability to generate realistic motions that are faithful to a multi-track timeline from the user.

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
