# OpenReview forum: "Multi-Track Timeline Control for Text-Driven 3D Human Motion Generation"
_thecvf.com/CVPR/2024/Workshop/HuMoGen — CVPR 2024 Workshop HuMoGen Submission_

### Official Review · Reviewer_uReN · 2024-03-28
**Temporal and Spatial control for text-driven motion synthesis, proposing a test-time denoising method integrated with pre-trained motion diffusion models.**

**Rating:** 4
**Confidence:** 5

**Review:**

On the bright side, the exposition is clear, and the results surpass baselines both quantitatively and qualitatively. Additionally, the MTT dataset makes a significant contribution to the community.

On the downside, the paper lacks novelty as it primarily combines SINC and DiffCollage. Additionally, the absence of metrics for individual sub-motions raises concerns about potential degradation in synthesis quality compared to previous art. Lastly, while the overall quality is high, there are occasional inaccuracies, such as motions not matching text prompts (e.g., lack of applause when mentioned).

Overall, I am inclined to accept this paper as it meets the standards of a workshop.

---

### Official Review · Reviewer_5Wdi · 2024-04-01
**This work addresses the problem of multi-prompt text-to-motion generation with temporal constraints using a pre-trained motion diffusion model. While the writing quality is decent and the proposed solution combines existing methods effectively, the significance of the work is limited due to unsatisfactory results.**

**Rating:** 2
**Confidence:** 4

**Review:**

**Summary**: This paper attempts to solve the problem of multi-prompt text2motion generation with temporal constraints per prompt using a pre-trained motion diffusion model. Similar to previous works, authors use the pre-trained motion diffusion model to generate a motion sequence per text prompt and its time-interval and then perform a form of stitching/blending to combine the generated motions to get a final cohesive motion sequence that adheres to all the provided conditions. Specifically, authors propose to use a temporal composition similar to DiffCollage and the spatial composition of SINC to combine the motions.

**Writing Quality**: Overall, the writing quality is decent, the problem definition seems clear and framework figures help clarify the method. However, I find the the method section a bit incomplete as certain details about different parts of the proposed solution are not clearly described. For instance, in section 3.2, it is said that body-part timeline conditioning is done following SINC but no further explanation is provided on how exactly SINC has done this in the method section.

**Results Quality**: I belive the weakest part of this paper is the results as it is very difficult to assess the significance of the proposed solution based on the provided experimental results. The quantitative experiments are missing relevant baselines such as PriorMDM which also performs motion stiching. Also, the final realism metrics are not satisfactory and show very minor improvement over the baselines. Qualitative results provided via the supplementary video also show defects. Firstly, some of the text prompts are completely ignored by the model. For instance, the very first motion in the video has "walking backwards" and "applaud" for one of the intervals and "appluad" is completely ignored. Since these two prompts are not inherently incompatible with each other, I do expect the generated motion to adhere to both of these conditions. As the main contribution of the paper is allowing for multiple prompts over a single time interval, these results are indicative of limitations in the model's ability to effectively incorporate multiple prompts within a single time interval, undermining the core contribution of the paper. Secondly, there are some artifacts in the provided results such as foot sliding and floating. Although physical artifacts are expected from motion diffusion models, the observed artifacts seem exacerbated by the addition of mutliple constraints.

**Clarity**: I strongly recommend improving the clarification of the method section in the later revisions. Apart from some missing details in the method section, I find the writing clear.

**Originality**: The problem of study is not necessarily original and the proposed solution is also not novel as it generates different motion sequences and applies a form of blending (as previously done before such as in PriorMDM) built on available methods applied to different problems (body-part composition of SINC and temporal composition of DiffCollage). However, I believe a simple and effective working solution is always appreciated, especially considering that it is not obvious how SINC and DiffCollage can be applied to this problem. Therefore, I believe authors have done a decent job making a working solution combining available methods and applying them to the problem of study.

**Significance**: Unfortunately, this work lacks significance as the provided results are not satisfactory.

---

### Meta-Review · Area_Chair_MytV · 2024-04-04

**Recommendation:** Accept

**Metareview:**

The paper addresses the task of combining temporal and spatial control for text-driven motion synthesis.

Pros:
* Well written
* a novel task definition
* A novel dataset
* Qualitative results generally exhibit high quality

Cons:
* Limited novelty
* Absence of quantitative assessment of individual sub-motions
* A small portion of the qualitative results does not match the text prompts (e.g., applause)

**Guidance to authors:**
* Please ensure that the new dataset is made publicly available
* Please incorporate quantitative results concerning individual sub-motions

---

### Decision · Program_Chairs · 2024-04-06

**Decision:**

Accept

**Comment:**

The paper will be published as part of the official CVPR workshop proceedings upon submission of the camera-ready version.